# Refined Simulation Method for Computer-Aided Process Planning Based on Digital Twin Technology

**DOI:** 10.3390/mi13040620

**Published:** 2022-04-15

**Authors:** Yupeng Xin, Yiwen Chen, Wenhui Li, Xiuhong Li, Fengfeng Wu

**Affiliations:** 1College of Mechanical and Vehicle Engineering, Taiyuan University of Technology, Taiyuan 030024, China; whalesss@163.com (Y.C.); lixiuhong@tyut.edu.cn (X.L.); 2College of Aeronautics and Astronautics, Taiyuan University of Technology, Taiyuan 030024, China; liwenhui@tyut.edu.cn (W.L.); wufengfeng@tyut.edu.cn (F.W.)

**Keywords:** digital twin, simulation, computer-aided process planning (CAPP), machining, manufacturing systems engineering

## Abstract

Simulation technology is widely used in computer-aided process planning (CAPP). The part machining process is simulated in the virtual world, which can predict manufacturing errors and optimize the process plan. Simulation accuracy is the guarantee of process decision-making and optimization. This article focuses on the use of digital twin technology to build a high-fidelity process model, taking the advantage of the integration of multiple systems, in order to achieve the dynamic association of real-time manufacturing data and process models. Making use of the CAPP/MES systems, the surface inspection data of the part is fed back to the CAPP system and associated with the digital twin process model. The wavelet transform method is used to reduce the noise of the high-frequency signal of the detection data, and the signal-to-noise ratio (SNR) is calculated to verify the noise reduction effect. The surface topography, after noise reduction, was reconstructed in Matlab. On this basis, the Poisson reconstruction algorithm is used to reconstruct the high-fidelity process model for the refined simulation of the subsequent processes. Finally, by comparing the two sets of simulation experiments with the real machining results, we found that the simulation results, based on the digital twin model, are more accurate than the traditional simulation method by 58%.

## 1. Introduction

The research on computer-aided process planning (CAPP) technology based on artificial intelligence has been widely developed [1]. In modern intelligent manufacturing systems, CAPP is no longer an independent system, but part of product life-cycle management (PLM) system, integrated with enterprise resource planning (ERP), the manufacturing execution system (MES), and other systems [2,3]. The boundary of the original definition of CAPP has been gradually blurred in practical application. Using artificial intelligence and big data analysis to realize dynamic process planning is the inevitable trend of CAPP technology in the future.

As is well-known that the two types of CAPP approaches are variant and generative, which are based on knowledge [1]. However, such knowledge-based approaches rely on statistical data or models. The real-time manufacturing data, collected in the actual manufacturing process, is separated from the theoretical model in CAPP stage. There is no mature modelling method and correlation mechanism that can merge the real-time manufacturing data and theoretical process model, which results in a process plan that cannot be dynamically updated and optimized according to the processing status of the workshop.

In the Industry 4.0 context, the statistical CAPP mode is not only difficult to effectively integrate with technologies, such as Internet of Things (IoT), big data, and artificial intelligence, but it also cannot meet the requirements of micro-processing and super-finishing for process digital models. Its disadvantages are mainly manifested in: (1) the in-process models cannot be directly used in digital measurement, which makes the CAPP and digital measurement systems unable to be integrated effectively; and (2) due to the lack of manufacturing data of the part, it is easy to cause a deviation between simulation and actual processing quality.

On the other hand, manufacturing machines and tools are increasingly equipped with sensors and communication capabilities, which have the ability to collect dynamic manufacturing information in real-time [4,5]. Advances in sensor and communication technologies provide foundations for linking the physical world of machines to the cyber world of computation. Such an integration and convergence into a cyber-physical world of manufacturing gives rise to a new focus on digital twin (DT). It provides a complete digital footprint of a physical system in the manufacturing process [6]. Decision-making and optimization of CAPP can then exploit these data, which can be updated in real-time on the physical system, through the synchronization enabled by sensors. DT technology can not only be used for modelling and simulating machining process, but it can also support operation and manufacturing services for optimized operations and failure prediction [7,8].

## 2. Related Work

The traditional CAPP system usually uses a series of drawings to express the geometric changes and processing requirements of the part. This mode is increasingly mismatched with today’s intelligent manufacturing system, which is mainly reflected in the following aspects: (1) the traditional CAPP system uses drawing files as the carrier, which cannot meet the needs of three-dimensional simulation, AR/VR, CNC machining, and 3D printing. (2) In manufacturing progress, one revision in a processing drawing will often lead to massive changes in the subsequent drawings. These modifications are geometrically relevant, but the drawing files are usually independent data sets and lack of correlation with each other. (3) Traditional CAPP systems mostly use knowledge to make process decisions and inferences, which belong to static process planning. In modern intelligent manufacturing systems, sensors, RFID, and other IoT technologies can achieve efficient integration between manufacturing units and systems, which makes it possible to use multi-dimensional, multi-level, real-time manufacturing data for dynamic process decision-making and optimization.

In order to make up for the shortcomings of the traditional CAPP system in data integration, process changes, and process decision-making, this article focuses on the application of DT technology in dynamic CAPP technology and refined simulation. This section summarizes the related works and progress of DT-based applications, in the field of machinery manufacturing, and analyzes the deficiencies and gaps of current research.

### 2.1. Digital Twin (DT)-Based Manufacturing Related Information Model, Framework, and Application Scenario

Combining Industry 4.0 concepts and imagining the future workshops, Tao proposed a concept of a DT workshop based on DT technology and discussed the related framework, system composition, operating mechanism, and key technologies [6]. Rainer proposed a “Digital Twin 8-dimension model”. On this basis, their team spent three years conducting social research (including experts from academia and industry), theoretical research (i.e., framework, modelling method, and future research directions of DT), and experimental verification on digital twin (building a prototype system of digital twin intelligent factory in the laboratory) [9]. Zhuang proposed a DT-based smart production management and control framework for the complex product assembly shop floor [10].

DT technology has a variety of application scenarios in the manufacturing field. In order to achieve in-process quality control of aerospace component manufacturing, Liu and Bao developed multiple DT sub-models, based on biomimicry principles [11]. These models comprised of an integrated true representation of the physical machining process. Liu proposed a CMCO (i.e., configuration design motion planning control development optimization decoupling) design architecture [8]. Through embedded PLC control and customized software development in a hollow glass smart manufacturing system, the interaction between networks and physical devices in different places (Chengdu and Guangzhou) is realized. Luo and Hu explored a mapping strategy between the cyber and physical worlds and established a multi-domain, unified modelling method of DT. On this basis, the method is verified by predicting and diagnosing the faults of CNC milling machine tool [12].

Judging from the literature in recent years, scholars have different understandings of the digital twin model and framework; however, they have reached a certain consensus, in some aspects, as below:

The digital twin model needs to have high fidelity, so that it can restore the real behavior and state of physical entities in the virtual world. Therefore, the DT information model often contains multiple dimensions [6,9].The interaction between virtual models and physical entities is the key to the role of DT technology. The real-time interaction directly affects the simulation results in the virtual world.Value-added services, obtained through data analysis, are the purpose of DT technology application. The manufacturing services that DT technology can provide for users mainly include equipment operation and maintenance [13], production scheduling optimization [14], processing/assembly precision control [10], and manufacturing energy consumption management [15].

### 2.2. Research Gaps

In recent years, digital twin technology has gradually become a research hotspot and received extensive attention from industry and academia. From the above research, we know that, in the virtual world, simulation based on the digital twin models can provide guidance, prediction, and optimization for the production activities in the physical world [7]. However, in the field of precision machining, there is a big gap between the simulation result and actual processing effect [16,17]. The following three aspects may be the main reasons:

The information contained in the model is incomplete and imprecise. The simulation model, in the process design stage, is an ideal model and lacks micro-scale surface quality information in the actual machining process [17].The assumptions and preconditions in the simulation are not accurate, and the model verification is insufficient. Simulation experiments based on ideal models usually require hypothetical conditions [16,17], which usually have a certain gap with the actual physical state.Simulation lacks an accurate method to evaluate the results. The evaluation method should be able to determine how the simulation experiment meets the target.DT technology emphasizes the integration of virtual and reality. Through the integration of cyber and physics, the real state in the physical world is restored to the greatest extent in the virtual world. In this paper, by studying the application framework of digital twin-based process planning (DTPP) technology, the authors explored a process planning mode for refined simulation using measured data on the surface of parts, in order to improve the accuracy of simulation results and provide more reliable predictive data for process decisions.

The rest of this paper is organized as follows. In Section 3, the information models of DTPP framework are highlighted. Consequently, Section 4 describes the framework of DTPP and proposes a new refined process simulation method based on DT in-process models. Following that, a case study is given to prove the proposed method. Finally, conclusions and identified areas for future research are given.

## 3. Digital Twin-Based Process Planning (DTPP) Information Model for Machining

Combined with digital twin technology, the concept and content of the process model should be newly defined and divided, which lays the foundation for the construction of the DTPP system.

**Definition** **1.**
*Digital twin-based process planning refers to the structured process digital models corresponding to the actual product manufacturing process. It consists of a series of three-dimensional virtual models that are completely consistent with related physical objects and can simulate the behavior and performance of real product manufacturing processes. The process information model of DTPP can be expressed as:*

(1)
$$DTPP=DTDM\cup \sum_{i=1}^{n}DTpm_{i}\cup DTEM$$

*where DTDM means to DT design model of product; DTpm_i_ means the ith in-process model; DTEM represent the equipment models, which mainly contain DT models of machine tool, cutting tool, and fixture.*


**Definition** **2.***Digital twin in-process model (DTPM) expresses the intermediate state model of part machining. During the machining process, the geometric dimensions and surface quality are constantly changing. This process is similar to biological evolution* [11]. * In order to truly depict the changing state of the part during machining, DTPM needs to constantly “evolve”, according to the machining conditions. From the perspective of virtual world, this “evolution” process can be described as the continuous superposition of multi-scale and -dimensional real-time manufacturing data on the as-designed model, so that it has the dual function of guiding the machining operation and feeding back the processing effect of the process. DTPM can be expressed as Formula (2):*(2)$$DTpm=Adm\cup \sum_{i=1}^{t}Step_{i}\cup Phy\;$$
*where Adm represents as-designed model, which includes geometric information, such as three-dimensional geometric dimensions, machining precision, etc. Step_i_ includes detailed processing step information of each operation, such as the cutting method, processing parameters, and tool path. The processing steps are organized in units of processing features, which can be expressed as:*
(3)$$Step=\sum_{i=1}^{n}Geo_{i}^{Step}\cup \sum_{j=1}^{m}Ano_{j}^{Step}\cup \sum_{k=1}^{s}Cam_{k}^{Step}\;$$
*where* $Geo_{i}^{Step}$
*means to geometric information of the machining feature corresponding to the processing step*. $Ano_{j}^{Step}$
*means to the accuracy requirements of the machining features.* $Cam_{k}^{Step}$
*represents the manufacturing information of the processing step, which mainly includes the cutting mode, cutting parameters, and tool path (G code or CAM post-processing program) information.*
*Phy means to physical state information of the part, considering the influence of materials, temperature, vibration, and other factors on the processing quality of parts. Phy needs to include not only surface quality inspection information, but also force, temperature, noise, vibration, and other information.*


**Definition** **3.**
*The digital twin equipment model (DTEM) includes the machine tool, fixture, cutting tool, presetter, disassembly device, AGV, and more. In order to reflect the dynamic processing information of equipments, DTEM should include the elements and behaviors related to the machining process of multiple types, time scales, and granularities. These contents are complex and numerous, and it is difficult to express them comprehensively through a single information model. Usually, different information models need to be constructed, according to different process types and application fields. This article establishes DTEM model for CNC machining as Formula (4):*

(4)
$$DTEM=em_{Geo}\cup Task\cup Cap\cup Mfg\;$$

*where*

$em_{Geo}$

*means to the three-dimensional geometric model of the process equipment, which is consistent with the actual physical size. Task means to the machining tasks, which reflects the current machining tasks of the equipment in real-time and is used for workshop scheduling and calculating of tool changing time. Cap represents the processing capacity information of the equipment, such as the maximum part size that can be processed by the machine tool, maximum diameter of the drill bit, and maximum clamping force of the fixture, thus providing a basis for selecting the process equipment and optimizing process scheduling. Mfg represents real-time manufacturing information of devices (machine tool, cutting tool, and fixtures) and includes processing time, personnel status, production progress, etc.*


The above information models can be divided into three categories, in terms of data expression and storage type: structured, unstructured, and semi-structured data. Structured data refers to row data, which is stored in a database and can be expressed logically using a two-dimensional table structure. The structured process data in DTPP models is composed of text, numbers, and characters, such as the contents of process operations and steps, machine tool information, tooling information, etc. Unstructured data refers to data that cannot be expressed using a two-dimensional logical table in the database, which mainly refers to the geometric element information in the three-dimensional solid model, real-time information (such as force, torque, and voltage) collected by the sensors, motion simulation animation, or AR file/picture. These data are generally composed of some independent data formats, and the data format files of this type are processed by special software. In addition, semi-structured data is between structured and unstructured, and it is composed of unstructured data and structured data or multiple structured data in a special form. The semi-structured data in DTPP models mainly refers to dimensional tolerance, geometric tolerance, and roughness, which are usually composed of special symbols and numbers.

According to different data type structures, a variety of information-expression methods need to be adopted, and the DTPP models shown in Figure 1 are taken as examples to illustrate. Structured data, such as process content “rough turning cylindrical surface” and other textual process information, is usually associated with the three-dimensional model, by means of three-dimensional annotation or attribute addition. Unstructured data is usually expressed by means of model format conversion and “model-data” association. For example, the tool path information needs to be converted from the CAD model to the CAM model. Real-time manufacturing data, in different formats, collected by sensors, can be associated with 3D models through intermediate format conversion files (such as XML) to provide data support for finite element analysis and machining motion simulation.

The DT process model uses the DT design model and DT in-process models as carriers, which can completely define the “static” process (geometric dimensions, surface roughness, positioning clamping, processing requirements, and others) and “dynamic” information (real-time data collected by sensors, workshop logistics information, machine tool status information, etc.). Each DT in-process model responds to each process operation and multiple processing step. Combined with model-based definition (MBD) technology, the DT in-process model can visually express the process operation or specify the machining area corresponding to the processing step.

The DT process model combines MBD, IoT, and other technologies to realize the association of manufacturing information (including “static” and “dynamic” process information) with the 3D solid model. Compared with the traditional process model, the DT process model can not only reflect the hierarchical structure relationship between the process route, process operations, and processing steps, but can also realize machine tool failure prediction, tool wear assessment, and workshop scheduling by analyzing real-time manufacturing data. Therefore, the scope covered by the process planning is extended to the production in the downstream workshop. The connection between process planning and workshop manufacturing is much closer.

## 4. DTPP Framework

As the carrier of DTPP, DTPM carries most of the information related to the process planning. The “static” process information and “dynamic” manufacturing data attached to DTPM advances the evolution (according to the time sequence), drives the generation of new DT models, and completes the process planning and iterative changes. Based on the geometric correlation between DTPM models, this paper proposes a DTPM-based process planning mode, as shown in Figure 2.

### 4.1. Process Route Design

In the process route design stage, the information of machining features and design requirements are obtained from the DT design model, and the top-level process route is designed by the knowledge database and an artificial intelligence algorithm. In the research of traditional CAPP technology, the decision and optimization of the process route mainly depends on the detailed summary and classification of process knowledge and relies more on manual experience. On the basis of summarizing the typical processes, new process routes are generated through human–computer interaction revision [1]. However, this method mainly establishes the association between the static machining feature and process knowledge. Since once a static machining feature is recognized, its associated process data is unchanged during the machining process.

In order to make up for this defect and improve the flexibility of process route design, DTPP technology considers the real-time operation status of the devices and overall resource scheduling of the workshop. Compared with the former, the arrangement of the processing sequence is more in line with the actual production conditions of the workshop. Therefore, the design of the process route, under the DTPP mode, needs to be integrated with CAM and MES systems. It can timely adjust the processing allowance and cutting parameters of the predetermined process, according to the detection data fed back by sensors, or adjust the processing route in real-time, according to the busy and idle status of machine tools. The workflow of DTPP system is shown in Figure 3.

The overall objective for this framework is to feed back dynamic DT data that change the process route by dealing with them via a fully automated method via a decision-making algorithm. The detailed workflow is as follows:

The process route, designed before product machining, is completed in the DTPP system. It takes the DT design model as input, analyzes the machining features, and uses the static process database and decision algorithm to design the process route.Whether this decision result can guide the actual production needs to be judged in conjunction with the dynamic DT data of the workshop. This process integrates DTPP, CAM, and MES systems. If the production conditions meet the processing requirements, the process command is executed by the devices. If not, the MES system feeds back the DT data to the DTPP system to modify the process route.In the MES system, it is divided into the physical and cyber layers, according to functions. The main function of the physical layer is to provide real-time data for the DT models and execute machining command. At the same time, the cyber layer is responsible for identifying error conditions and predicting equipment failure [18].

In Figure 3, we can grasp the product processing status and equipment working status in real-time and use “dynamic” process data to improve DTPP’s flexible design capabilities. In this case, it is particularly important to build digital twin models that can reflect the state of the product and devices in the process.

### 4.2. DT In-Process Models Design

During the machining process, the machining features and surface quality of the part are constantly changing. In order to truly depict this changing process, the DT in-process model needs to constantly “evolve”, according to the processing process. From the perspective of digital manufacturing, this “evolution” process can be described as the continuous superposition of multi-scale and -dimensional real-time manufacturing information (which is defined as *Phy* in Definition 2) on the as-designed model, so that it has the dual function of guiding the processing operation and feeding back the processing effect of the process. According to the chronological order of process planning and part processing, this information superposition process is divided into two stages: the digital definition stage before processing (pre-processing stage) and real-time manufacturing information superposition stage during processing (in-processing stage).

Pre-processing stage

The digital definition before processing includes establishing an ideal three-dimensional geometric model of the process, defining the processing size and accuracy requirements, selecting processing methods and devices, and planning processing steps. The processing steps include: determining the processing area, setting cutting parameters, and planning the tool path. The role of this stage is to guide the actual processing operation. The three-dimensional geometric model of the process carries all the static process information related to the processing operation. The DTPP system uses the process knowledge base and decision rules to automatically select the processing parameters and devices. The CAM system provides the processing steps of the NC program for tool cutting.

2.In-processing stage

The as-designed model in the virtual environment cannot truly reflect the real topography changes during the processing of parts. Normally, researchers obtain inspection data through measuring instruments to reflect the real effects of parts after processing. However, the inspection data is not related to the as-designed model. The three-dimensional model in the virtual environment still expresses the ideal processing effect. The errors generated during the processing are not supervised and controlled in the virtual environment. Therefore, in order to truly map the microscopic changes of the surface topography of the parts in the physical world, the machining inspection system needs to be integrated with the virtual simulation system to feed back the machining inspection data to the virtual world. On this basis, DTPM in the virtual world needs to undergo modal changes. According to the Delaunay triangulation theory [19], the as-designed model evolves into a discrete point cloud or triangular mesh model, corresponding to the location of the inspection point.

This evolution process is shown in Figure 4. The part is measuring online, and the measurement data should be fed back to an intelligent controller, which is composed of a comparator, knowledge library, rule library, and memory. The comparator contains two inputs and one output. *DTpm_i−1_*, before processing, provides the desired dimension as one input and measurement data as another input. The comparator feeds the processing error to adaptive compensation module, which is composed of a knowledge library, memory, and rule library. The knowledge library contains the processing parameters (such as spindle speed, feed rate, and cutting depth) corresponding to different processing conditions. The memory stores the historical processing parameters relating to same parts, providing big data samples for deep learning. The rule library adjusts the processing parameters according to the change of machining allowance.

The results of compensation contain the dimensional value and modified processing parameters, which will feed back to the cyber and physical words. In the cyber world, using MBD technology, the DTPM will be given new size information or processing parameters after the inspection process. In the physical world, the modified cutting parameters are fed to servo motor driver, thus adjusting the cutter moving for the next procedure operation.

In the above feedback mechanism, DTPM can be applied to the application criteria required by digital twins; that is, the information required for processing is related to physical entities, multivariate data is integrated with digital twin models, dynamic real-time interaction is compatible with digital twin models, and the requirements of the multi-system integration applications is matched with the digital twin information model [20].

### 4.3. Refined the Machining Simulation, Based on Digital Twin In-Process Model (DTPM)

From the perspective of expressing the surface data of the part, DTPM, which is continuously updated through the above feedback mechanism, has a higher fidelity and is closer to the actual physical part. Therefore, in the abstract world, DTPM can be used not only for the simulation of CNC machining programs but also for deeper refined simulation. The former is to use the principle of image science to correct the numerical control program, and the latter is to simulate the process mechanism using scientific calculations, such as materials science, heat transfer theory, solid mechanics, and fluid mechanics, to judge the feasibility of the process and predict the processing effect. Refined simulation strives to simulate the surface morphology of the part, close to the real machining state, from the mesoscopic or microscopic scale, so as to achieve the purpose of replacing physical experiments.

Therefore, the infinite description is required to be able to consider all kinds of geometric deviations from a macro- to a nano-scale and capture these different variations [21]. It is not possible to clearly define all of the geometric specifications of the machining process of part. However, a finite description can be used for calculations and simulations, such as assembly analysis or tolerance simulations, for predicting the subsequent manufacturing processes. In general, the finite descriptions of DTPM can be wire frames, point clouds, surface meshes, volume models, and cell models. The point cloud representation method has many advantages in the precise expression of the geometric deviations of the part. Furthermore, point coordinates can be obtained by 3D scanners in the manufacturing inspection applications. This provides the possibility to improve the accuracy of the simulation results. The flowchart of this procedure is shown in Figure 5, as follows.

However, for a point cloud model, its measurement data will be very huge, even if it is an extremely small size model. In order to reduce the workload of the calculation, it is necessary to filter out unnecessary measurement data and extract points that will affect the subsequent simulation results. For this purpose, it is necessary to model deviations for the finite point cloud model. Different approaches for modelling deviations can be found in the literature, such as wavelet analysis, which is used to filter out white noise in scanning [22]. The commonly used filtering methods mainly include Fourier transform-based filtering [23,24], one-dimensional wavelet transform filtering [25], and Bayesian post-processing [26,27]. In general, the 1D Gaussian and muti-Gaussian methods are considered to be able to reasonably sample points from point cloud model to establish random deviation model [26,28].

Based on the real-time measurement data, the reconstructed DTPM is closer to the actual surface of the part. This model is used to replace the ideal model in the DTPP system, which makes the simulation results in the virtual world more authentic and instructive.

### 4.4. DTPP-Integrated System Framework

In order to realize the above functions, this paper proposes a DTPP-integrated application framework, as shown in Figure 6.

DTPP extends the scope of the traditional process design by linking process planning to the real-time execution of the shop floor using technologies such as IoT/RFID. The DTPP framework, shown in Figure 6, includes six layers. The meaning and content of each layer are as follows:**application layer** DTPP’s application range includes process design, tooling design, machining, assembly, and inspection. Its potential applications are the real-time optimization of process routes/parameters, tool life prediction, equipment status monitoring, processing plan evaluation, processing quality control, and more.**service layer** The biggest advantage of DTPP is to make full use of real-time industrial big data to achieve dynamic process planning. In order to meet computational efficiency and reduce the cloud computing load, telecom standards organizations and operators are studying how to deeply integrate with mobile internet and IoT services in future 5G networks, thereby increasing the value of mobile network bandwidth. Mobile edge computing (MEC), proposed by the European Telecommunications Standards Institute, is a technology based on 5G evolution architecture and deeply integrating mobile access networks with Internet services [29]. On the one hand, MEC can improve the user experience and save bandwidth resources. On the other hand, by sinking computing power to mobile edge nodes, it provides third-party application integration, which provides unlimited possibilities for service innovation of the mobile edge portal. The seamless integration of mobile networks and applications will provide a powerful weapon for dealing with various DTPP applications.**smart devices****layer** Closed-loop control is the main feature that distinguishes the DTPP system from the traditional CAPP system. The hardware that performs information feedback mainly includes various types of sensors, RFID tags, and communication equipment. Therefore, the DTPP system is no longer a pure process design software system, but an integrated application system that combines software and hardware.**system layer** DTPP requires the synergistic response of multiple systems, including product design, process design, manufacturing, inspection, operation and maintenance systems, etc. Each system needs to be open and easy to integrate between different systems.**platform layer** The DTPP platform refers to the software operating platform used by process personnel when designing the process. In addition to the process operation function of the traditional CAPP system, the platform also integrates the real-time process simulation function based on the digital twin model. For the machining process, simulation mainly includes the real-time simulation of mechanical equipment (machine tools, tools, and fixtures) and pre-simulation of the parts.**technology layer** The technology supporting the above applications can be divided into two parts: model-based process planning and digital twin technology. Model-based process planning technologies support process route planning, machine and cutting tool selections, tool path planning, etc. Digital twin technology functions mainly focus on real-time big data acquisition, processing, and analysis. DTPP has higher requirements for communication technology. Currently, 5G technology defines the next three application scenarios: enhance mobile broadband (eMBB), massive machine type of communication (mMTC), and ultra-reliable low latency communications (uRLLC). In the future, with the support of 5G technology, it is possible to make DTPP easier to implement.

The dynamic process design mode is the main difference between the DTPP system and traditional CAPP system; the scope of process knowledge is further expanded (including real-time manufacturing data), and the simulation technology is easy to exert greater advantages. In addition, the DT models, contained in the DTPP system, are also an important part of the digital twin workshop. The DTPP system can provide more accurate real-time manufacturing data for the workshop’s APS advanced planning, logistics, and warehousing.

## 5. Cased Study

This section presents an implementation of prototype DTPP system. Section 5.1 uses this framework to build a DTPP prototype system in the Cyber physical system (CPS) environment of a machining workshop. Section 5.2 presents a comparative experiment of refined simulation.

### 5.1. A Prototype System of DTPP

According to the above framework, a DTPP prototype system, based on the workshop CPS system, has been designed. Figure 7 shows the CPS system interface, DT equipment model, and machine tool in the IoT environment. The DTPP system is a secondary development of the NX8.5 system using Visual Studio 2010. As a part of the Siemens PLM system (TeamCenter10 (TC)), it is integrated with the CPS system to realize dynamic process design, based on DTPM.

Based on the real-time status data of the workshop equipment, acquired by the CPS, a digital twin process model is constructed through the transmission of the processing data between the different systems, and its implementation is shown in Figure 8. The CNC machine tool adopts the Shenyang i5 intelligent CNC milling machine and Jiangsu Dongqing CNC wire cutting machine DK7732; additionally, an open EtherCAT bus is used for real-time data transmission between the master control system. It can receive DTPM geometry model from NX8.5 system through PLM/MES integration. The detection instrument is the ZeGagage Plus optical profiler from the ZYGO company. Since the instrument does not have networked data transmission, the detection data is manually entered in the experiment, and the data is stored in the Oracle database file and fed back to the DTPP system to achieve refined simulation. A standard communication protocol, such as OPC UA, is used between the heterogeneous control and DTPP system to realize the transmission of field data.

Considering the timeliness and consistency of product data transfer, the manufacturing process planner (manufacture structure editor (MSE)) module and DTPP module in TeamCenter are used to build the process structure synchronously. The synchronization function of DTPP and MSE ensures the consistency of data updates. The system realization scheme is shown in Figure 9.

Process designers obtain process design tasks in TC, which include: part models, processing requirements, real-time status information of workshop and equipment, and so on. In order to realize the automatic acquisition and update of DTPM model objects, the DTPP navigator and process route planning interface are established by using the MFC framework of Visual Studio 2010 and NX/Open (an application programming interface provided by NX to customize and extend NX). Using model-based definition (MBD) technology to define DTPM models in NX, these models are ideal in the process design stage.

### 5.2. A Case Study of Refined Simulation Based on DTPM

In order to verify the superiority of DTPM in refined processing simulation, this paper takes the simulation of the barrel finishing process of a satellite part, as shown in Figure 10, as an illustrative example. After the part is processed by heat treatment and drilling, the cross groove in the center of the part and 12 circumferentially distributed through-hole grooves are processed by wire EDM, with a groove width of 0.8 mm. Because, after wire EDM, the cutting boundary and surface of the part will be burnt and damaged, as shown in Figure 10c. The last process of the process plan uses the barrel grinding method to polish the part. The EDEM discrete element simulation method is used to predict the effect of barrel grinding. By comparing the simulation results of DTPM and the ideal model, the superiority of the refined simulation method based on DTPM is verified.

Surface sampling

For a point cloud model, even a small area contains a huge number of points. On the premise that the physical experimental results corresponding to the simulation can be observed, the sampling area of this experiment is 1 square millimeter, and the three-dimensional shape data of a 1024 × 1024 lattice is obtained. Using Matlab to visualize the measurement data in three dimensions, the results are shown in Table 1.

2.DTPM reconstruction

The wavelet transform method [24] is used to denoise the measured data. The selected wavelet basis function is db1, three-layer wavelet decomposition is performed, and soft threshold is selected for calculation. The threshold is selected using adaptive Stein’s unbiased risk estimation principle [29], and the threshold is calculated by Matlab software to obtain the threshold thr = 0.0478.

The obtained 1024 × 1024 three-dimensional space sampling point sequence is expanded to obtain 1024 sets of lateral height data for denoising processing. Firstly, the original height signal is flattened to obtain 1,048,576 data points of one-dimensional signal data, which is reconstructed by wavelet decomposition. The reconstruction process is shown in Figure 11, where S is the initial signal, a_3_ is the high-frequency signal after the decomposition of the third layer, and d_1_, d_2_, and d_3_ are the low-frequency signals of the first, second, and third layers, respectively. Compared with the original signal, the denoised, one-dimensional signal data can clearly show that the denoised signal eliminates outliers, as shown in Figure 12. Calculating the signal-to-noise ratio (SNR) before denoising is 31.57 dB, and the proportion of noise signal is 0.69%, which is less than 1%. It can be considered that the denoising did not damage the topographical features of the original surface.

Through the above steps, a 3D visualization model of the sample area can be obtained, as shown in Figure 13b. However, such a model has no closed boundary and cannot be directly applied to simulation software to simulate the physical processing process. In order to establish a DTPM that can be used for refined simulation, the obtained surface height matrix is used to reconstruct the three-dimensional process geometric model, using the Poisson reconstruction algorithm in MeshLab. The set parameters are shown in Table 2.

The basic idea of Poisson reconstruction is to convert the discrete point information of the surface topography into a continuous, integrable surface function, thereby constructing an implicit indicator function surface, derived from the object, and the normal vector of the point cloud represents the inner and outer directions [30].

The Poisson reconstruction process of the surface model is shown in Figure 13. Import the original surface topography data after denoising, that is, the three-coordinate data file, shown in Figure 13b, into Meshlab. The normal vector of the discrete point cloud on the surface is calculated in Meshlab to obtain the 3D point cloud data with vectors, and then the Poisson reconstruction function is used to construct the surface model, as shown in Figure 13c. The generated surface is exported to Obj format, through the Meshlab software; the Boolean sum operation is performed on the exported 3D surface topography sheet model and 3D model of the part in the UG/NX software, and the result (shown in Figure 13d) is obtained.

3.Simulation in EDEM

The EDEM discrete element simulation software is used to simulate the machining process of the example part in the BJL-LL05 vertical centrifugal barrel polishing and finishing equipment. The machining principle is shown in Figure 14. The part, media, abrasives, water, etc., are loaded into the drum. The four drums are evenly distributed in the base and move in a planetary motion. The revolution speed is *N*, rotation speed is *n*, and speed ratio is n/N. The media achieves surface finishing by collision, rolling, and micro-grinding on the surface of the part [31].

The Hertz–Mindlin with the Archard wear model was selected as the contact model for analysis in EDEM. The wear constant is $1\times 10^{-10}\;{\mathrm{Pa}}^{-1}$ (select the alumina media with an average diameter of 3 mm), filling rate is 70%, barrel speed is 300 r/min, speed ratio is −1, and revolution radius is 135 mm. Set the simulation time to 30 s, in which the media is generated within 0~1s, and the barrel makes planetary movements in 2~30 s. The time step is set to 20% of the Rayleigh time, and the calculated time step is $\Delta t=1.5\times 10^{-5}\;s$. Taking the polishing and finishing of a single barrel as an example, the processing parameters are shown in Table 3.

The wear depth cloud diagram, with a time interval of 5 s, was selected to analyze the wear of the part. The wear results of the ideal surface model and DTPM in the EDEM simulation are shown in Figure 15 and Figure 16. Comparing the two sets of pictures, it can be seen that Figure 15 reflects the disordered scratches of the media on the ideal surface. On the other hand, the scratches in Figure 16 reflect a certain regularity.

4.Simulation results

In order to analyze the effectiveness of the simulation results, the simulated grayscale images of t = 30 s in the two sets of experiments were compared with the actual processing results, and the results are shown in Figure 17.

The similarity between Figure 17a,c, denoted as *SIM_1_*, and *SIM*_2_, represents the similarity between Figure 17b,c. The similarity value is obtained by calculating the normalized correlation coefficient of the two images [32], as shown in Formula (5).
(5)$$SIM=\frac{\sum_{m}\sum_{n}\left(A_{mn}-\bar{A}\right)\left(B_{mn}-\bar{B}\right)}{\sqrt{\left(\sum_{m}\sum_{n}{\left(A_{mn}-\bar{A}\right)}^{2}\right)\left(\sum_{m}\sum_{n}{\left(B_{mn}-\bar{B}\right)}^{2}\right)}}$$
where *A_mn_* and *B_mn_* are the value of the *m*th row and *n*th column in the gray value matrix A and B of the two images. $\bar{A}$ and $\bar{B}$ are the average of the pixels of the matrix.

Using Formula (5), *SIM*_1_ = 0.0128 and *SIM*_2_ = 0.5986 are obtained. That is, compared with the ideal surface model, DTPM is more in line with the wear situation in actual machining, which verifies the advantages of DTPM in the refined simulation.

## 6. Conclusions and Future Work

This paper proposed and validated a novel DTPP mode, based on digital twin technology. Some of the contributions of this research are listed below.

A process information model, based on digital twin technology, is proposed. Its main innovation is that it takes DTPM as the process object and integrates the processing process information and real-time machining information into the three-dimensional model.A new DTPM reconstruction method is proposed. The surface topography data, measured during the processing, is attached to the ideal 3D model surface; on this basis, the refined processing simulation is realized.A set of simulation comparison experiments, based on the ideal model and DTPM, were designed. The simulation results of the two were compared with the actual processing effect, and the superiority of refined simulation based on DTPM was verified.

The research in this article also has some shortcomings. It can be seen from the comparison of simulation results in Figure 16 that, although the result of *SIM*_2_ was much higher than that of *SIM*_1_, *SIM*_2_ was much smaller than 1, which means that the simulation results based on DTPM still have a large gap with the actual results. The main reason for these results is that, due to the huge amount of simulation calculation, the processing simulation time was set to 30 s, and the simulation calculation time was 48.354 h. The simulation computer was an Intel i7-10700 processor with a main frequency of 2.9 GHz and 32 GB memory; the EDEM simulation used a CPU with 16 cores. The processing time of the actual part is 5 min, so the value of *SIM_2_* is much less than 1.

In order to make up for this shortcoming, in future research work, we consider adopting simplified sampling points and combining mathematical statistics to process simulation calculation data, so as to achieve the purpose of shortening the simulation calculation time.

## Figures and Tables

**Figure 1 micromachines-13-00620-f001:**
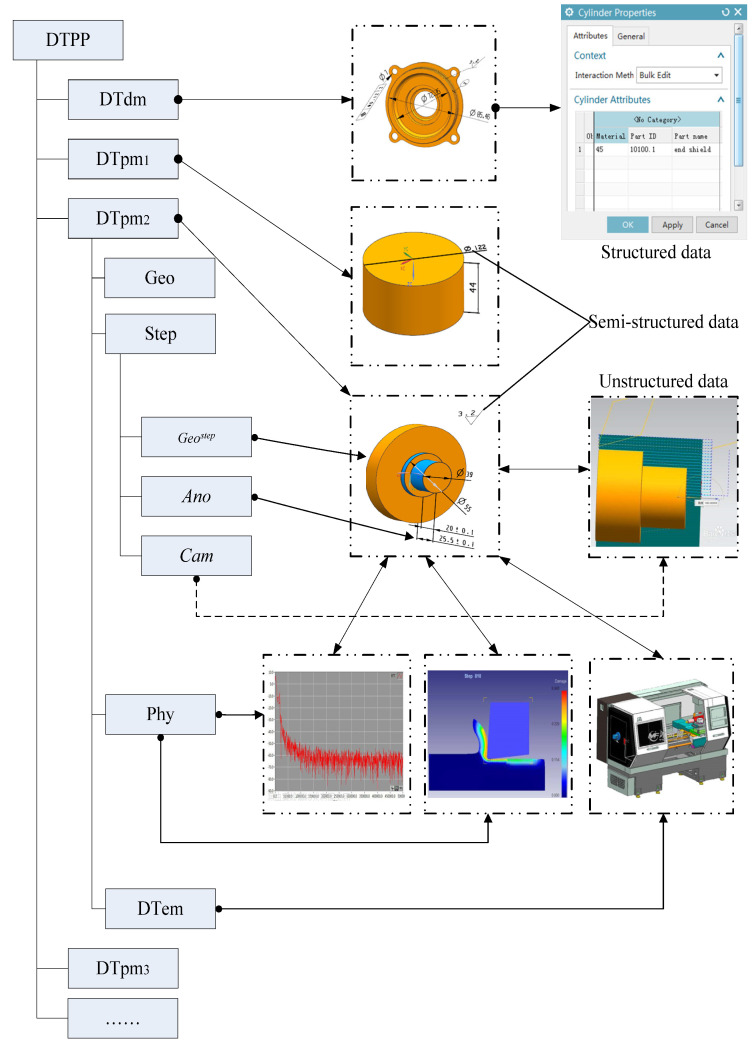
A sample of digital twin (DT) process models.

**Figure 2 micromachines-13-00620-f002:**
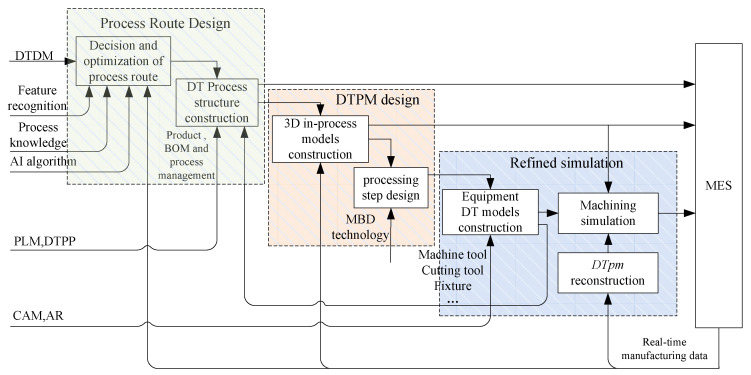
A Digital twin in-process model (DTPM)-based process planning mode.

**Figure 3 micromachines-13-00620-f003:**
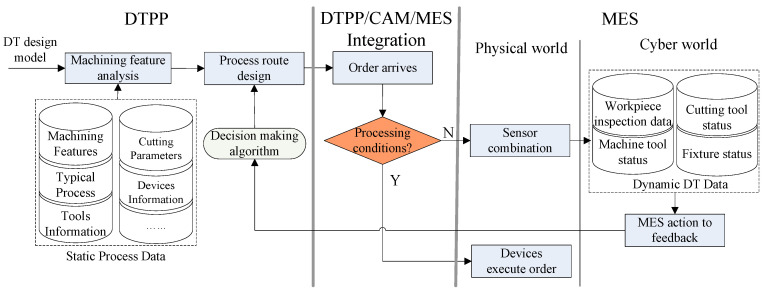
The workflow of digital twin-based process planning (DTPP) system.

**Figure 4 micromachines-13-00620-f004:**
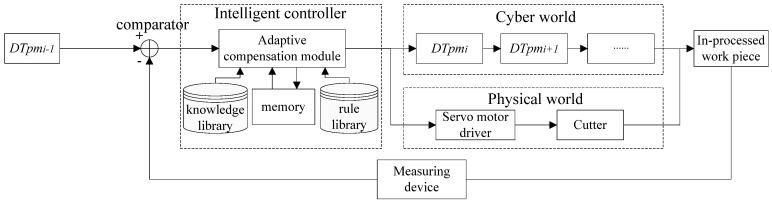
The reconstruction mechanism of DTPM.

**Figure 5 micromachines-13-00620-f005:**
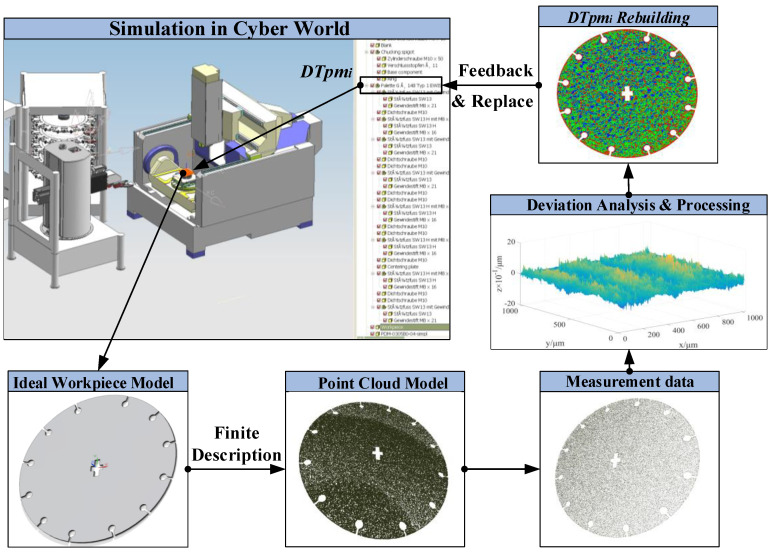
The finite element representation and feedback of DTPM.

**Figure 6 micromachines-13-00620-f006:**
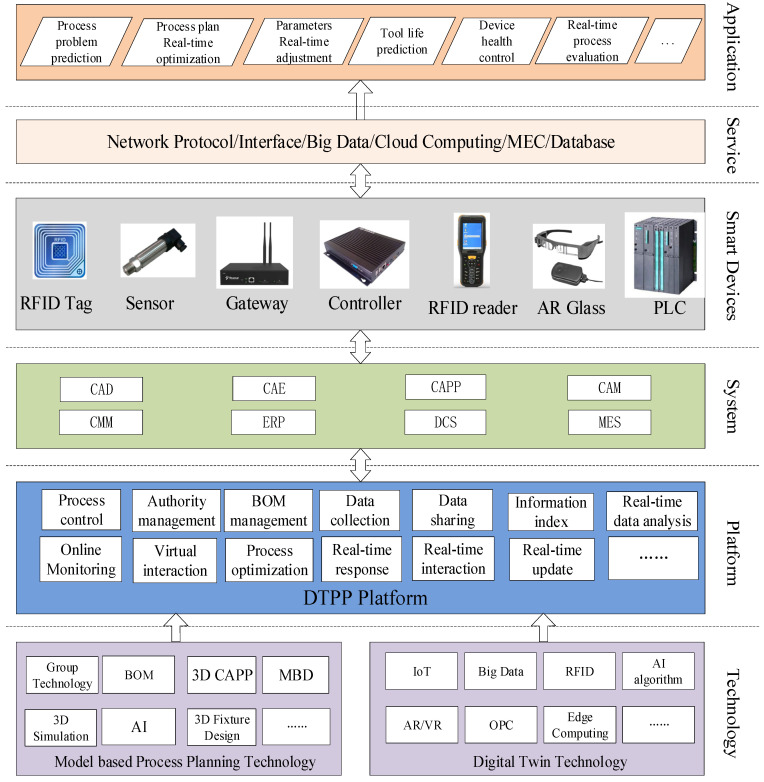
The framework of DTPP system.

**Figure 7 micromachines-13-00620-f007:**
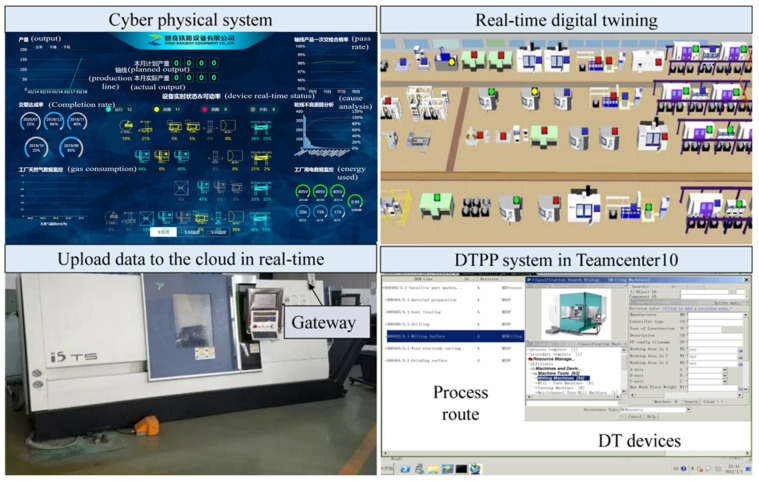
The Cyber physical system (CPS) environment of workshop.

**Figure 8 micromachines-13-00620-f008:**
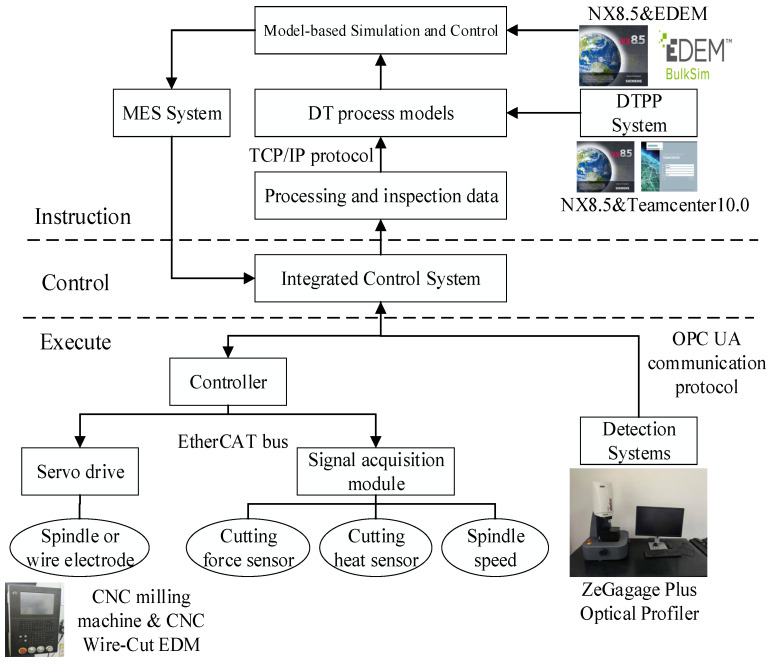
DTPP system implementation scheme and data flow.

**Figure 9 micromachines-13-00620-f009:**
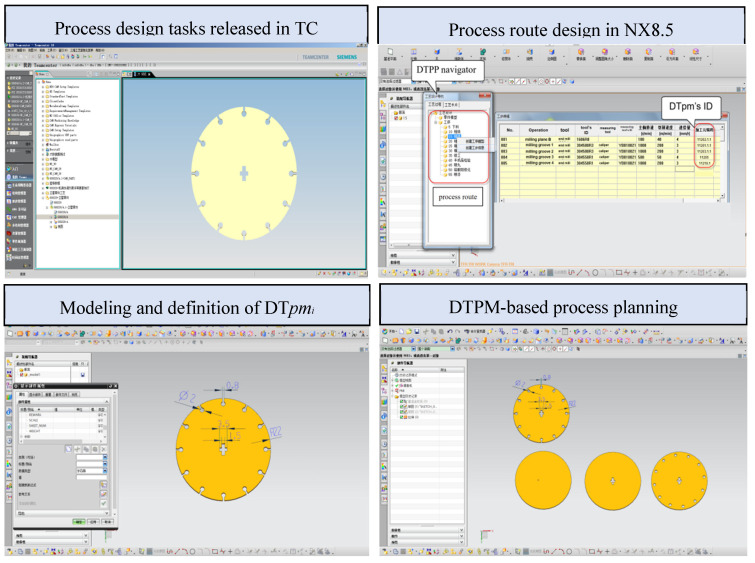
DTPM-based process planning in NX8.5.

**Figure 10 micromachines-13-00620-f010:**
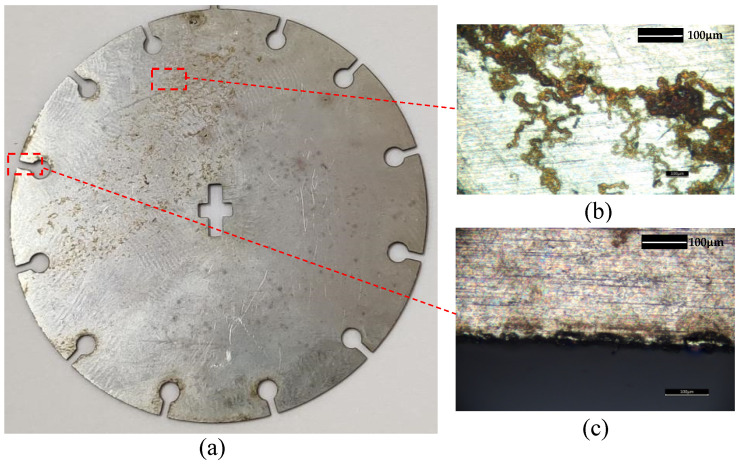
Damaged state of the part before finishing. (**a**) Part before finishing. (**b**) Surface corrosion. (**c**) Edge burning.

**Figure 11 micromachines-13-00620-f011:**
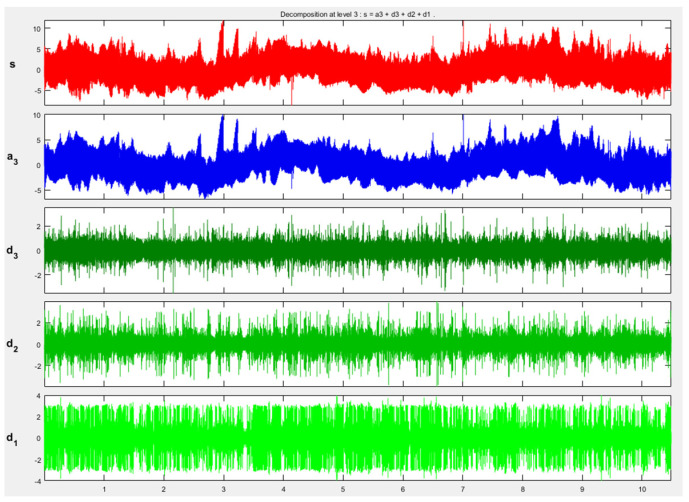
Wavelet transform results of 3D surface topography data.

**Figure 12 micromachines-13-00620-f012:**
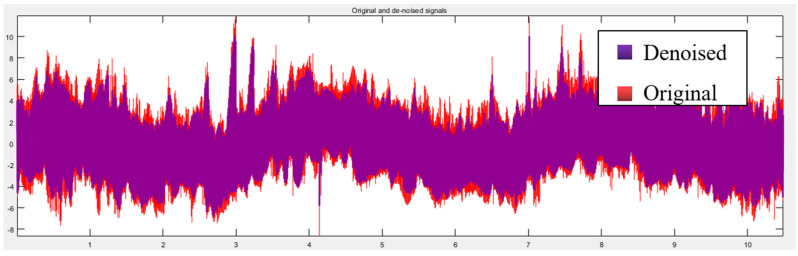
Comparison results of noise reduction.

**Figure 13 micromachines-13-00620-f013:**
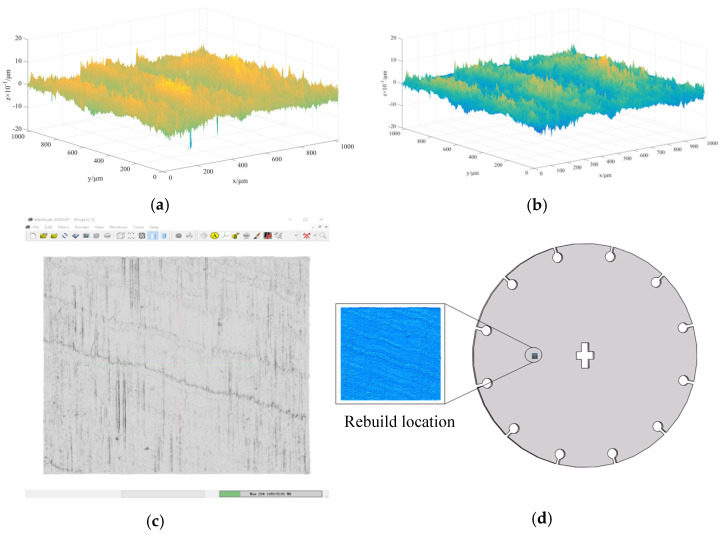
The reconstruction process of DTPM. (**a**) Surface before noise reduction. (**b**) Surface after noise reduction. (**c**) Poisson reconstruction in Meshlab. (**d**) Surface after Poisson reconstruction.

**Figure 14 micromachines-13-00620-f014:**
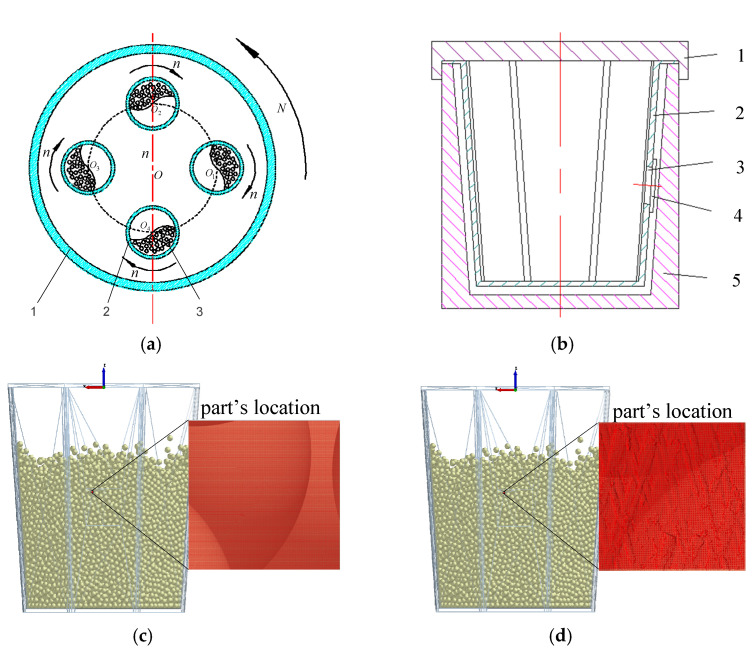
Schematic diagram of centrifugal barrel finishing. (**a**) 1. Pedestal; 2. barrel; 3. media. (**b**) 1. Barrel head; 2. inner barrel; 3. part; 4. silicone pad; 5. barrel. (**c**) Simulation using ideal model. (**d**) Simulation using DTPM.

**Figure 15 micromachines-13-00620-f015:**
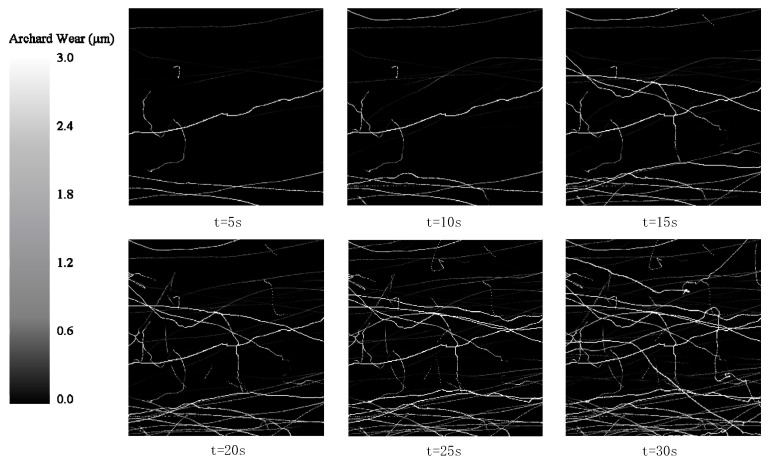
Cloud map of wear depth of ideal model.

**Figure 16 micromachines-13-00620-f016:**
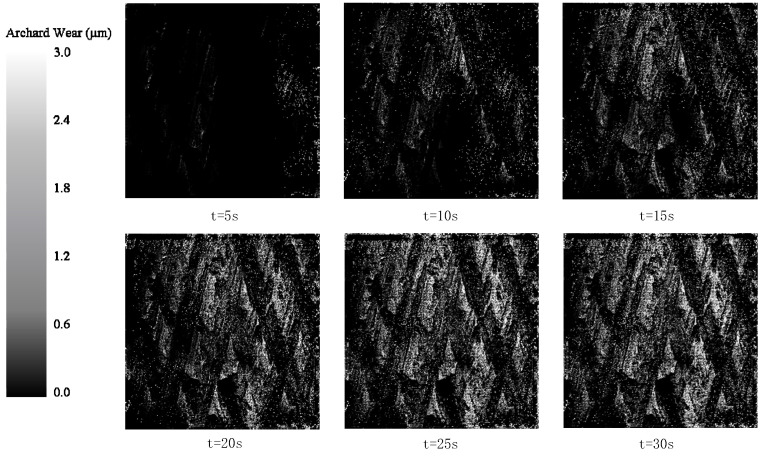
Cloud map of wear depth of DTPM.

**Figure 17 micromachines-13-00620-f017:**
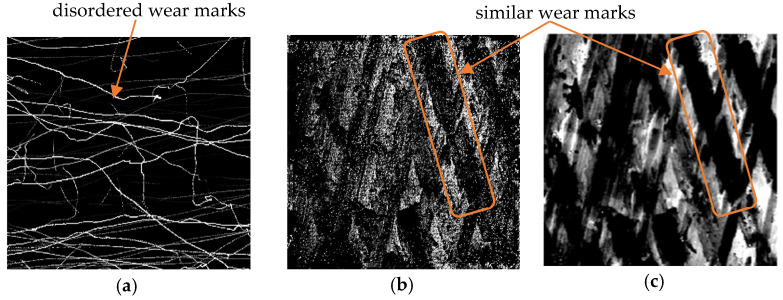
Comparison of simulation results and actual results. (**a**) Ideal model-based simulation. (**b**) DTPM-based Simulation. (**c**) Actual wear results.

**Table 1 micromachines-13-00620-t001:** Example of 3D coordinate data of part surface topography.

Reconstruction of 3D Topography in Matlab	*Z* (μm)	*X* (μm)	*Y* (μm)
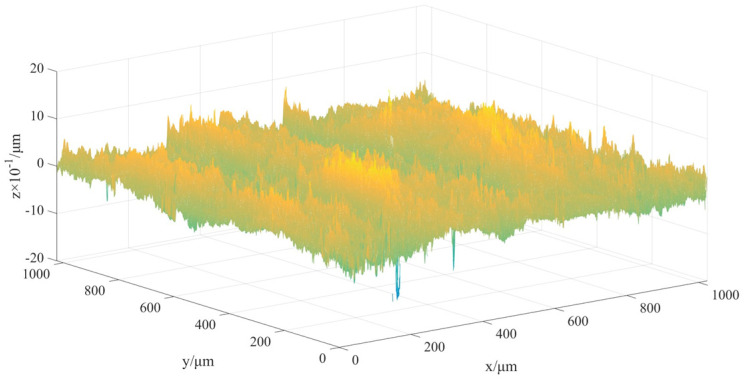	1.046103 × 10^−1^	0	0
	1.250018 × 10^−1^	0.814815	0
	1.065271 × 10^−1^	1.629630	0
	1.438948 × 10^−1^	2.444445	0
	1.140261 × 10^−1^	3.259260	0
	1.114373 × 10^−1^	4.074075	0
	6.302840 × 10^−2^	4.888890	0
	6.135895 × 10^−2^	5.703705	0
	9.818698e × 10^−2^	6.518520	0
	1.2412273 × 10^−1^	7.333335	0
	1.4751230 × 10^−1^	8.148150	0
	2.0934136 × 10^−1^	8.962965	0
	···	···	···

**Table 2 micromachines-13-00620-t002:** Setting parameters of shielded Poisson reconstruction.

Structural Depth/μm	Adaptive Octree Depth/μm	Conjugate Gradient Depth/μm	Scale Factor	Minimum Number of Samples	Interpolation Weight	Gauss-Seidel Relaxation Factor
10	5	0	1.1	1.5	4	8

**Table 3 micromachines-13-00620-t003:** The main parameters of the sample model in the EDEM simulation.

	**media**	**part**	**barrel**
material	${\mathrm{Al}}_{2}O_{3}$	Aluminium alloy	Photosensitive resin
poisson ratio	0.36	0.33	0.4
elastic modulus/Pa	1.26 × 10^7^	2.632 × 10^10^	9.246 × 10^8^
density/**(** $\mathrm{kg}\cdot m^{-2}$**)**	2675	2700	1150
	**m** **edia–media**	**p** **art–media**	**b** **arrel–media**
modulus of resilience	0.35	0.5	0.35
coefficient of static friction	0.15	0.45	0.3
coefficient of kinetic friction	0.46	0.15	0.15

## Data Availability

Not applicable.

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
