# Peer review of "Refined Simulation Method for Computer-Aided Process Planning Based on Digital Twin Technology"

_micromachines, 2022, doi:10.3390/mi13040620_

Round 1
Reviewer 1 Report
The proposed virtual system is made based on a virtual system suggested by Zhuang. Its better to work based on a real system as in concepts of the industry 4.0.
Many tools used in the proposed system, its such like a miss in the paper contents. Its not clear, how all those tools will affected on CAPP using this proposed system.
Author Response
Point 1: The proposed virtual system is made based on a virtual system suggested by Zhuang. Its better to work based on a real system as in concepts of the industry 4.0.
Response 1: Thank you very much for this question raised by the reviewer, which gave us new thinking about the article.We have added a detailed description of the system implementation in Section V, "Cased study". As described in the first paragraph of 5.1 "The DTPP system is a secondary development of the NX8.5 system using Visual Studio 2010. As a part of the Siemens PLM system (TeamCenter10, TC), it is integrated with the CPS system to realize dynamic process design based on DTPM." A new second paragraph describes how these systems interface with the hardware. Figure 9 shows how to use DTPM for process design in the NX8.5 environment.
Point 2: Many tools used in the proposed system, its such like a miss in the paper contents. Its not clear, how all those tools will affected on CAPP using this proposed system.
Response 2: Indeed, as stated by the reviewer, too many tools and equipment are included in the DTPP system framework. The original text not only does not explain in detail how they work on CAPP, but the relationship between them is also logically problematic. We first sorted out the logical problems of the DTPP system framework, and revised Figure 6 and the corresponding content in the original text; secondly, Figure 8 was newly added in the fifth part "Cased study", which explained the DTPP system implementation scheme and data flow in detail.

Reviewer 2 Report
This work focuses on defining a framework that exploits the digital twin technology to achieve the dynamic association of real-time manufacturing data and process models. As such, it is a very interesting work, thought the following are my main concerns:
1°) The formalization proposed shows that the framework is based on a series of static data (that are represented by static models) catching the on-going reality of the manufacturing elements within the manufacturing process. However, it is not clear how all these dqta/static models are translated into some dynamic model related to the manufacturing process itself. Therefore, it gives the impression the framework makes use of snapshots as data collected from the shop floor, but the reflection of these data on the dynamic simulation model(s) are not obviously formalized.
2°) Authors make too much use of "et al.". The paper is full of such terms, and it would be good the corresponding sentences be reformulated differently.
3°) The framework presented in Figure 6 looks akward to me, as the models layer comes between the service layer and the devices layer.
Author Response
Point 1: The formalization proposed shows that the framework is based on a series of static data (that are represented by static models) catching the on-going reality of the manufacturing elements within the manufacturing process. However, it is not clear how all these data/static models are translated into some dynamic model related to the manufacturing process itself. Therefore, it gives the impression the framework makes use of snapshots as data collected from the shop floor, but the reflection of these data on the dynamic simulation model(s) are not obviously formalized.
Response 1: As the reviewer stated, the original article lacks a detailed description of how the data from the manufacturing process is applied to the simulation. We have added Figure 8 to Section V, illustrating how data from the manufacturing process flows to the simulation environment and enabling dynamic model reconstruction in both NX8.5 and EDEM simulation environments.
Point 2: Authors make too much use of "et al.". The paper is full of such terms, and it would be good the corresponding sentences be reformulated differently.
Response 2: We are very sorry for such a problem, and thank the reviewer for being able to find it during the review process.We carefully sorted out the whole text, adopted the methods of deleting and replacing synonyms, and changed a total of 15 "et al".
Point 3: The framework presented in Figure 6 looks akward to me, as the models layer comes between the service layer and the devices layer.
Response 3: Admittedly, there is a logical problem with Figure 6, and we are very sorry for that. We have re-discussed the composition of the DTPP system framework, modified Figure 6, and modified the corresponding content. First of all, we believe that the service layer is for applications, and smart devices are the carriers that provide manufacturing services. Based on this consideration, the top-down order of Figure 6 is adjusted to the application layer, the service layer, and the smart device layer, respectively. On this basis, the hardware should have supporting software systems, and these systems should be integrated and compatible, which constitute the basis of the DTPP system. Therefore, after the smart device layer is the system layer and the DTPP platform. Finally, the technologies that support the realization of various functions of DTPP are model-based process planning technology and digital twin technology.
